# Wnt/β-Catenin Signaling and Immunotherapy Resistance: Lessons for the Treatment of Urothelial Carcinoma

**DOI:** 10.3390/cancers13040889

**Published:** 2021-02-20

**Authors:** Alexander Chehrazi-Raffle, Tanya B. Dorff, Sumanta K. Pal, Yung Lyou

**Affiliations:** Department of Medical Oncology & Experimental Therapeutics, City of Hope Comprehensive Cancer Center, Duarte, CA 91010, USA; achehraziraffle@coh.org (A.C.-R.); tdorff@coh.org (T.B.D.); spal@coh.org (S.K.P.)

**Keywords:** Wnt, β-catenin, urothelial cancer, immune checkpoint inhibitor, immunotherapy resistance

## Abstract

**Simple Summary:**

Metastatic urothelial cell carcinoma (UCC) is a significant public health burden with a median survival estimated at about 15 months. The use of immunotherapy with immune checkpoint inhibitors has greatly improved outcomes but only benefits a minority (~20%) of patients. In this review we discuss the evidence showing how a key molecular pathway known as Wnt/β-catenin signaling can be a driver of immunotherapy resistance and how these insights can serve as lessons for improving future treatment of urothelial carcinoma.

**Abstract:**

Urothelial cell carcinoma (UCC) is a significant public health burden. It accounts for approximately 90 percent of all bladder cancers with an estimated 200,000 annual deaths globally. Platinum based cytotoxic chemotherapy combinations are the current standard of care in the frontline setting for metastatic UCC. Even with these treatments the median overall survival is estimated to be about 15 months. Recently, immune checkpoint inhibitors (ICIs) have demonstrated superior clinical benefits compared to second line chemotherapy in UCC treatment. However only a minority of patients (~20%) respond to ICIs, which highlights the need to better understand the mechanisms behind resistance. In this review, we (i) examine the pathophysiology of Wnt/β-catenin signaling, (ii) discuss pre-clinical evidence that supports the combination of Wnt/β-catenin inhibitors and ICI, and (iii) propose future combination treatments that could be investigated through clinical trials.

## 1. Introduction

Urothelial cell carcinoma (UCC) is the most common malignancy of the urinary system. It accounts for approximately 90 percent of all bladder cancers with an estimated 200,000 annual deaths globally [1,2]. UCC is also an aggressive histology as 25% of patients who receive potentially curative treatment for localized disease will unfortunately succumb to tumor metastasis.

Cytotoxic chemotherapy is the current standard of care in the frontline setting for metastatic UCC. The median overall survival is estimated to be about 15 months with modern chemotherapy regimens containing platinum-based agents [3,4]. Once patients progress on first line chemotherapy treatments the second line chemotherapies have limited efficacy with median progression-free survival periods of 3–4 months (Figure 1) [5,6].

More recently, immune checkpoint inhibitors (ICIs) have demonstrated superior clinical benefits compared to second line chemotherapy in UCC treatment [7,8]. However, only a minority of patients (~20%) respond to ICIs in the treatment of UCCs and other malignancies [7,8,9]. It has also been noted that those patients who respond to ICIs can often maintain an impressive durable response lasting more than 14–15 months [7,8]. This phenomenon has been observed across numerous cancer subtypes [10], highlighting the need to better understand the mechanisms behind ICI resistance.

Several mechanisms of ICI resistance in cancers have been reviewed extensively elsewhere [11,12]. Previously proposed resistance pathways include PTEN, FGF, MYC, TGFB, TP53, WNT, VEGF, and ANG2 [11,12]. The majority of studies investigating immunotherapy resistance mechanisms have been done in non-UCC studies; as a result, the proposals in this review extrapolate data derived from both urothelial and non-urothelial studies. In this review, we will (i) examine the pathophysiology of Wnt/β-catenin signaling, (ii) discuss pre-clinical evidence that supports the combination of Wnt/β-catenin inhibitors and ICI, and (iii) propose future combination treatments that could be investigated through clinical trials.

## 2. Canonical Wnt Signaling

Wnt signaling is a highly coordinated and conserved signaling cascade that occurs at the cell surface and within the cytoplasm. This pathway mediates an array of biological functions, including cell fate decisions during embryonic development, stem cell equipoise, and immune system homeostasis [13,14,15,16]. Recent reviews published elsewhere provide a more exhaustive discussion on the β-catenin dependent and independent pathways [17,18,19,20,21,22]. For the purposes of this review (which is most relevant to ICI resistance) we will focus primarily on β-catenin-dependent Wnt signaling.

Canonical, or β-catenin-dependent, Wnt signaling is one of the primary sources of dysregulated transcription in cancer. In the “on-state”, the signal cascade begins at the cell surface with Wnt ligands binding to the Frizzled:LRP5/LRP6 receptor complexes, and culminates in the nucleus with the formation of a transcription-activating complex [23]. The primary mediator of this cell surface-to-nucleus signal is β-catenin, a membrane/cytoplasmic armadillo repeat protein which lacks the ability to independently promote DNA transcription [17,20,24]. Instead, β-catenin is trafficked into the nucleus to DNA-binding T-cell factor (TCF)/lymphoid enhancer binding factor (LEF) transcription factors [24,25].

Once bound to DNA by TCF/LEFs, β-catenin recruits other co-activators and regulatory components that collectively activate transcription of the downstream genes known as the Wnt target genes. These sets of Wnt target genes drive cells to proliferate, self-renew, differentiate and survive in a variety of tissues and contexts. In normal cells, feedback inhibition results in this activity occurring only transiently, which in turn prevents overactivation of Wnt target gene transcription. Signal transduction is thus “turned off” in cells with low or absent Wnt because β-catenin becomes unstable by being tagged in the cytoplasm for ubiquitination by the destruction complex, which then leads to proteasome degradation.

However, in various cancers (i.e., colon cancer) mutations in the destruction complex components (e.g., *APC*, *AXIN2* and *FAM123B/WTX*) or regulators of the receptors/ligand (e.g., *RNF43/ZNRF3*, *RSPO2,* or *RSPO3*) components can lead to unchecked Wnt signaling. These mutations negate the cytoplasmic feedback controls and create cells with constitutive, high levels of β-catenin and aberrantly high levels of Wnt target gene transcription that can initiate carcinogenesis and immune suppression [20,26,27,28,29].

## 3. Upregulation of Wnt/β-Catenin in Bladder Carcinogenesis

Several correlative studies have shown conflicting evidence between upregulation of Wnt/β-catenin signaling and UCC carcinogenesis [30,31,32,33,34]. For instance, The Cancer Genome Atlas (TCGA) Research Network detected Wnt signaling alterations in 73% of UCC tumors [35]. However, Ahmad et al. noted Wnt signaling in only 33% of their clinical UCC samples [36,37,38]. The discrepancy could most likely be due to comparisons using different methods and patient populations. For example, one Ahmad et al. study used a tissue microarray array with core biopsy samples, whereas a TCGA study detected aberrations in Wnt signaling through genomics using RNA-seq and whole exome sequencing [35,36,37,38]. Also there was a difference in sample size with TCGA and Ahmad et al. studies using 131 and 60 patient samples respectively [35,36,37,38]. Additionally it was noted that β-catenin expression’s correlation to tumor grade and muscle invasion has been inconsistent [39]. Despite these discrepancies between studies, it is evident that a substantial proportion of UCC develops in the context of Wnt signaling aberrations.

From a pathophysiologic perspective, numerous pre-clinical studies have implicated the silencing of endogenous Wnt inhibitors as potential oncogenic events. CpG hypermethylation of the WIF1 (Wnt inhibitory factor-1) promoter was found to lead to decreased transcription and increased Wnt signaling activity in human bladder cancer cell lines [40]. Knockdown of WIF1 by siRNA in bladder cancer cell lines led to increased activity in c-myc and cyclin D1 mRNA transcription and increased cell growth [40]. These results suggested that Wnt signaling via WIF1 could potentially promote development of UCC [40]. Another proposed mechanism involves aberrations in the oncogene activation-induced cytidine deaminase, which upregulates the Wnt/β-catenin pathway and thereby promotes UCC growth [41]. More studies are needed to better understand how Wnt signaling can drive urothelial carcinogenesis.

## 4. Wnt/β-Catenin Induces Immune Cell Exclusion in Urothelial Cancer

Due to the limited efficacy of ICI treatments, much effort is being dedicated to developing predictive biomarkers of response and understanding the biological mechanisms for resistance. One widely established predictive biomarker for ICI response is intratumoral enrichment of CD8+ T-cells prior to treatment [42,43]. Therefore, many studies have used the presence and quantity of CD8+ T-cell infiltration as a surrogate marker when performing correlative studies to determine if other molecular pathways may be involved in predicting the ICI response.

A recent study by Sweis et al. used a bioinformatics approach to correlate CD8+ T-cell infiltration with various signaling pathways [44]. The investigators analyzed the whole exome sequencing (WES) and RNA-seq transcriptional profile data from the 267 samples of urothelial bladder cancer collected for the TCGA study. The investigators stratified these tumors based on a 160-gene T-cell inflamed expression signature indicative of a T-cell inflamed and non-inflamed microenvironment. This T-cell inflamed gene signature was then validated by performing immunohistochemistry (IHC) staining for CD8+ T-cell infiltration on a sample of 19 tumors (7.1%).

Once stratified into inflamed vs. non-inflamed phenotypes, the investigators uncovered that 730 genes were preferentially expressed in the non-T-cell-inflamed tumors. Ingenuity pathway analysis then showed that one of the top upstream regulators for these groups of differentially expressed genes were those that were regulated by β-catenin/Wnt signaling. The authors then went back to the 19 samples which they had initially performed CD8+ T-cell IHC staining and co-stained for nuclear β-catenin as a marker for active β-catenin dependent Wnt signaling. The investigators found a statistically significant inverse relationship between nuclear β-catenin and the density of CD8+ T cells infiltrating the tumor.

To further validate the Wnt signaling pathway as a mediator of non-T-cell-inflamed tumor microenvironments, a follow up study done by Luke et al. employed a similar approach (WES genomics and RNA-seq transcriptional profiling) and analyzed 9244 samples across 31 different types of cancers [45]. The investigators used their previously developed 160-gene T-cell inflamed expression signature to segregate the samples into T-cell inflamed, intermediate, or non-T-cell inflamed. The investigators defined Wnt/β-catenin signaling activation at three different levels: assessment of somatic mutations or copy number alterations in *CTNNB1* (gene for β-catenin) and other regulatory genes predicted to result in pathway activation, expression of downstream Wnt target genes, and β-catenin protein levels which were assessed through reverse phase protein array (RPPA). With respect to the 363 UCC samples included in this cohort, all three levels correlated with a non-T-cell inflamed tumor signature, the most pronounced of which was CTNNB1 protein level. Taken together, these findings suggest that there is a significant correlation between upregulation of Wnt signaling and a non-T-cell-inflamed microenvironment in UCC.

## 5. ICI Attenuation via CCL4

As previously discussed, translational studies have suggested that Wnt/β-catenin signaling may induce a non-T-cell-inflamed tumor phenotype thereby excluding immune cells from the tumor microenvironment and dampening the therapeutic effect of ICIs. To elucidate molecular mediators, the Gajewski group used a genetically engineered melanoma mouse model with active β-catenin signaling (BRAF/PTEN/CAT-STA) in the tumors [46].

In their mouse model, the authors found that β-catenin signaling activation was associated with low levels of tumor infiltrating CD8+ T-cells. Conversely, mice in which β-catenin signaling was absent contained a high density of CD8+ T-cell infiltration. In order to discern if this was due to differences in neo-antigens, the authors introduced a neo-antigen (SIY) expressing construct genetically into the tumors and adoptively transferred T-cells with SIY T cell receptor. They found that the transferred T-cells accumulated in the BRAF/PTEN-STA tumors but not the β-catenin expressing BRAF/PTEN/Bcat-STA tumors despite both tumors now expressing the neo-antigen. Furthermore, anti-PD-1 and anti-CTLA-4 agents were rendered ineffective in the Wnt-activated (BRAF/PTEN/Bcat-STA) mice but remained effective in Wnt-inactivated (BRAF/PTEN-STA) mice. These results suggested that upregulation of Wnt/β-catenin may indeed induce resistance to immune checkpoint inhibition.

The investigators then queried whether this blunted response to ICIs could be dependent on antigen presentation from CD103+ dendritic cells (DC). Within Wnt/β-catenin-activated T-cell-depleted tumors, they found that CD103+ DCs were nearly absent and IFN-β cytokine expression was reduced. The investigators then found that intratumoral injection of CD103+ DCs led to restoration of T-cells infiltration within the tumor. This supported the role of CD103+ dendritic cells as key mediators of an antitumor immune response. To characterize the mechanism of failed recruitment of the CD103+ DCs, the investigators analyzed the gene expression of these two tumor types and found that four chemokines (CCL3, CXCL1, CXCL2, and CCL4) were lower in the non-T-cell inflamed BRAF/PTEN/Bcat-STA tumors. Of these four chemokines, only CCL4 was found on an in vitro DC migration assay to possess the ability to effectively modulate cell migration.

Furthermore, the investigators found that the Wnt signaling target gene ATF3–which also binds at the promoter region of the CCL4 gene—was expressed at higher levels in the β-catenin activated melanoma tumors. This negative feedback was substantiated by then demonstrating that gene knockdown of ATF3 and CTNNB1 in melanoma cell lines led to upregulation of CCL4 expression (Figure 2).

## 6. Wnt/β-Catenin Signaling Induces Immune Cell Exclusion by Affecting the Tumor Microenvironment (TME)

Tumor-associated macrophages (TAMs) are amongst the most common tumor immune infiltrating cells in the tumor microenvironment (TME) [47]. TAMs are classically thought to exist in two polarized states with the activated M1 and M2 subtypes [47]. The M1 subtypes are thought to play a significant role in the anti-tumor immune response by producing reactive oxygen species (ROS) and pro-inflammatory cytokines [47]. The M2 subtype has been found to have an opposite immunosuppressive function by producing anti-inflammatory cytokines (i.e., IL1, IL-13, and TGF-β) which can promote tumor growth and ICI resistance [47]. These anti-inflammatory cytokines and chemokines can also induce the production of regulatory T-cells which directly inhibit cytotoxic T cells further driving immunosuppression [47,48,49].

It has been shown that Wnt/β-catenin signaling can modulate the TAMs population in the TME leading to a protumoral phenotype which may be ICI resistant [50,51]. In a study done by Kaler et al. using isogenic colon cancer cell lines (HCT116 and Hke-3 cells) with mutated active β-catenin, the investigators found that TAMs could further enhance the pre-existing Wnt/β-catenin signaling present and protect the cancer cells from TRAIL-induced apoptosis [50]. In contrast, when HCT116 cancer cells with an inactive β-catenin allele were cultured with TAMs, the investigators noted that these cells were susceptible to TRAIL induced apoptosis and were unable to increase their Wnt/β-catenin signaling levels [50]. The investigators also found that the isogenic colon cancer cell lines (HCT116 and Hke-3 cells) with mutated active β-catenin when cultured with TAMs would produce more snail protein, which is a known Wnt/β-catenin signaling target gene and driver of tumor mesenchymal transition [50]. These results suggested that the increased Wnt/β-catenin signaling from the TAMs could induce snail gene expression and drive a tumor mesenchymal transition phenotype [50]. Of note this nail driven tumor mesenchymal transition has recently been reported to be a possible mechanism for ICI resistance [50,51,52].

Another potential mechanism for ICI resistance is through the tumor’s ability to create a hostile TME that is acidic from increased lactic acid production which can lead to impaired cytotoxic T-cell function [53,54,55]. A detailed discussion on how tumors create a hostile hypoxic and acidic TME which leads to suppression of the T-cells’ cytotoxic function is beyond the scope of this manuscript. For a more comprehensive review on this topic there are many excellent reviews which can be found in the reference section of this manuscript [53,56,57]. Briefly, the oncogenic mutations that drive carcinogenesis (i.e., Akt/PI3k/mTOR and Wnt/β-catenin signaling) have also been shown to drive a metabolic reprogramming of cells from oxidative phosphorylation towards aerobic glycolysis [56,58]. This phenomenon wherein cancer cells prefer to undergo the more inefficient aerobic glycolysis even in the presence of oxygen has been known for almost 100 years since it was first described by Dr. Otto Heinrich Warburg [57,59]. It is thought that cancer cells have evolved this shift towards aerobic glycolysis as a way to produce metabolic byproducts which can then be converted to provide the needed biomass to use as building blocks for its rapid cell proliferation [56,57]. As the tumor grows larger in size its metabolic demands also increase in an unregulated manner which often outstrips the local oxygen and nutrient supply [53,56]. This imbalance in metabolic demand and available supply of local resources creates a hostile TME that is hypoxic, acidic (due to lactic acid build up), and nutrient deficient [53,56]. In addition, to the existing overactive oncogenic signaling pathways present in the cancer cells (i.e., Akt/PI3k/mTOR and Wnt/β-catenin signaling) these hostile TME conditions will further drive the tumors to adapt by increasing angiogenesis and glycolysis via the VEGF and HIF signaling pathways [53,56]. These same acidic and hypoxic TME conditions will then inhibit the oxidative phosphorylation that is needed by T-cells in order to perform their cytotoxic functions potentially leading to immunosuppression and ICI resistance [53]. In fact it has been shown that high lactate concentrations in the TME can impede the CD8+ T-cells ability to export lactate and suppress their natural cytotoxic function [60].

As discussed above, the Wnt/β-catenin signaling pathway was found to initially play a central role in carcinogenesis by driving cell proliferation [20]. More recently, it has also been found to play an additional role in cancer metabolism by metabolically reprogramming cancer cells to promote aerobic glycolysis and lactic acid production [58,61,62,63]. In a study done by Pate et al. the investigators found that by using genetically engineered human colon cancer cell lines that overactive Wnt/β-catenin signaling drives aerobic glycolysis and lactic acid production by upregulating the genes pyruvate dehydrogenase kinase 1 (PDK1) and monocarboxylate transporter 1 (MCT1/SLC16A1) [58,61]. They also found that when this metabolic shift towards glycolysis occurred that there was also a corresponding inhibition in the gene expression of pyruvate dehydrogenase (PDH) and oxidative phosphorylation [58,61]. Other independent studies have also provided further supporting evidence that Wnt/β-catenin signaling can drive the metabolic reprogramming of cancer cells towards lactic acid production and aerobic glycolysis [62,63].

It has also been shown that the lactic acid in the TME can play a role in immunosuppression and drive further tumor growth [53,54,64]. In a study done by Brand et al. the investigators found that patients who had melanoma tumors with increased LDHA gene expression and lactic acid levels were more likely to have findings of impaired T and NK cell infiltration consistent with an immunosuppressed or immune deficient tumor phenotype [54]. The investigators then used shRNA to create LDH_low_ murine melanoma and pancreatic cancer cell lines [54]. Through the use of various clever control experiments the investigators showed that knockdown of the LDHA gene resulted in a stable tumor cell phenotype that produced low levels of lactate with no effects on the other metabolic pathways analyzed [54]. They then proceeded to inject these murine melanoma and pancreatic cells lines which were either LDH_high_ or LDH_low_ into syngeneic mice [54]. They found that the LDH_low_ had impaired tumor growth and higher T-cell and NK cell infiltration compared to the LDH_high_ tumors [54]. These findings suggested that the acidic TME created by uncontrolled lactate production led to impaired immunosurveillance and T-cell and NK cell infiltration leading to an immune deficient TME [54]. In another independent study done by Harel et al. the investigators found that increased oxidative phosphorylation and lipid metabolism in melanoma tumors by proteomic analysis were more likely to have potentiated antigen presentation and response to anti-PD1 immune checkpoint inhibitor or TIL-based immunotherapy [64]. The investigators of the Harel et al. study concluded that the tumors with increased oxidative phosphorylation were undergoing less glycolysis, secreting less lactate, and creating a more favorable TME for immune cells [64].

The above studies provide evidence supporting the hypothesis that the presence of lactic acid in the TME can be immunosuppressive by inhibiting the needed oxidative phosphorylation of cytotoxic T-cells. As a result, this has led to the proposal that targeting lactic acid production could be a potential way to overcome ICI resistance [55]. In summary, the above findings provide evidence that Wnt/β-catenin signaling can drive ICI resistance by modulating the TME through the interaction with TAMs or driving lactic acid production and creating a local immunosuppressive environment for cytotoxic T-cells (Figure 3) [50,53,54,55,56,57,58,61,64].

## 7. Overcoming ICI Resistance with β-Catenin Inhibition

The above mentioned studies provide strong evidence that the Wnt/β-catenin signaling pathway drives immune cell exclusion which can then lead to immune checkpoint inhibitor resistance in cancer treatment [44,45,65]. As a result, one could reason that combining a Wnt/β-catenin signaling inhibitor and ICI may lead to overcoming this resistance mechanism (Figure 4).

Early therapeutic efforts primarily centered on finding targets for Wnt inhibition [23,66,67,68]. However, one of the major hurdles that researchers encountered was developing a molecule small enough to penetrate the nuclear membranes yet robust enough to counteract the large β-catenin regulatory complex [23,68,69]. Another challenging adverse class effect was on-target bone toxicity, which ultimately led to the early termination of several Phase I studies [23,68,70,71,72].

More recently, several studies have shifted focus toward downstream inhibition of the intranuclear transcriptional β-catenin complex to enhance immune cell infiltration within the tumor microenvironment. For instance, Ganesh et al. designed a β-catenin inhibitor (DCR-BCAT) that selectively silenced CTNNB1 (the gene which transcribes/β-catenin) in tumors using an RNAi oligonucleotide [73]. Using allografted B16F10 mouse melanoma cells on immunocompetent C57BL/6 mice, which are known to be refractory to ICI treatments through T-cell exclusion [73], Ganesh et al. found that treatment with DCR-BCAT significantly increased the intratumoral density of CD8+ T-cells compared to the placebo control. Quantitative analysis of tumor RNA detected a decrease in β-catenin gene expression as well as a concomitant increase in CCL4 expression. Furthermore, single-cell flow cytometry of the DCR-BCAT mouse tumors showed a significant increase in CD8+, CD3+, CD103+, and PD-1 positive cells, suggesting that these tumors were transitioning to a T-cell-inflamed phenotype.

Encouraged by these results, the investigators subsequently examined if their β-catenin inhibitor could reconstitute an immune response within their T-cell-excluded tumor model. Although monotherapy with either the DCR-BCAT or an ICI was minimally effective, the combination of DCR-BCAT plus ICI elicited a synergistic effect with reductions in tumor size by as much as 87% [73]. Moreover, the authors confirmed that this combination was effective in another model, the Neuro2A (neuroblastoma) cell lines, which are also non-T-cell-inflamed at baseline [73]. These findings suggest that a β-catenin inhibitor can effectively downregulate Wnt/β-catenin signaling and induce a T-cell-inflamed phenotype that can potentiate a response to immune checkpoint inhibitors [73].

## 8. Ongoing Clinical Trials and Future Directions

In recent years, several novel agents with varied mechanisms of action have attempted to mitigate the immunosuppressive tumor microenvironment through WNT/β-catenin inhibition (Figure 5, Table 1). One such therapeutic effort in development is DKN-01, an antibody that antagonizes the WNT/β-catenin pathway through inhibition of DKK1 [74]. Preliminary results from a Phase 1b/2a study of DKN-01 plus pembrolizumab (NCT02013154) demonstrated a disease control rate of 80% in patients who had tumors with high DKK1 expression as compared to a disease control rate of 20% in patients with low DKK1 expression [74].

Another class of WNT/β-catenin inhibitors disrupt PORCN, an enzyme that facilitates WNT secretion [75]. A recent Phase I study of the PORCN inhibitor WNT974 combined with the PD-1 monoclonal antibody spartalizumab (NCT01351103) reported impressive results across several solid tumors, including stable disease in 53% of patients who were previously refractory to ICIs [76]. Of note, neither one of these trials included urothelial carcinoma and focused on other malignancies such as GI cancers, melanoma, and NSCLC. However, seeing how the combination of ICIs with WNT/β-catenin inhibitors has produced some signal of efficacy even in the early phase clinical trials, this combination warrants further investigation for the treatment of UCC.

## 9. Conclusions

In summary, WNT/β-catenin signaling can drive immune cell exclusion and may be a resistance mechanism for immune checkpoint inhibitors. Several preclinical studies have shown that inhibition of the WNT/β-catenin pathway in conjunction with an ICI can effectively overcome this resistance mechanism. With respect to UCC, this combination is particularly promising given the high frequency of WNT/β-catenin aberrations in correlative studies as well as its potential role in upregulating urothelial oncogenesis. Thus, to complement ongoing clinical trials across other solid tumors, additional studies that validate the synergistic relationship of ICIs and WNT/β-catenin inhibitors in UCC are urgently needed.

## Figures and Tables

**Figure 1 cancers-13-00889-f001:**
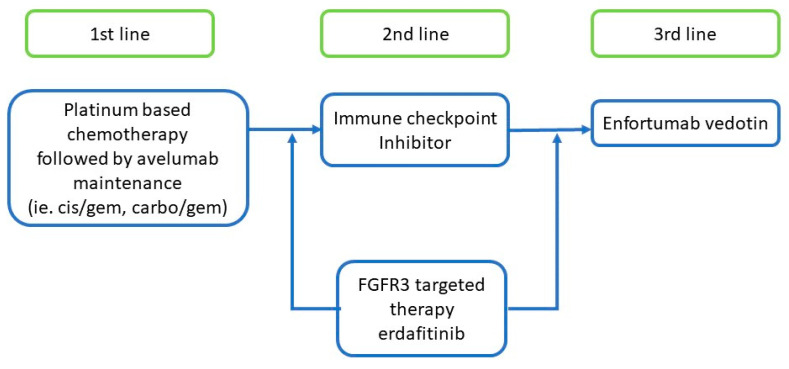
Current systemic treatments in metastatic urothelial cell carcinoma.

**Figure 2 cancers-13-00889-f002:**
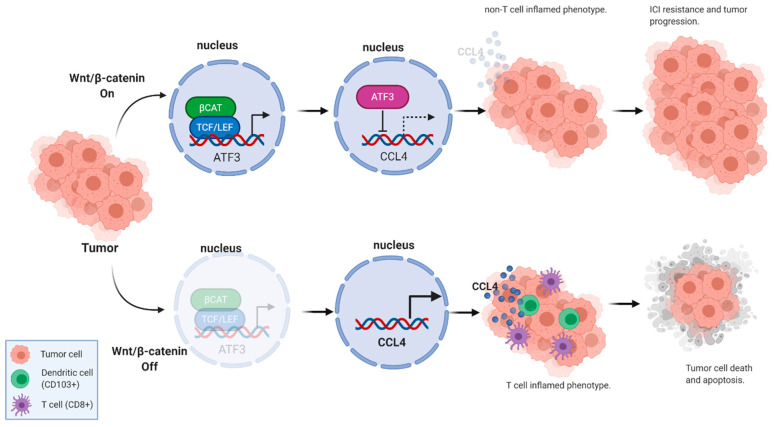
Wnt/β-catenin signaling can alter T-cell infiltration status and ICI response via CCL4. (Created with BioRender^®^).

**Figure 3 cancers-13-00889-f003:**
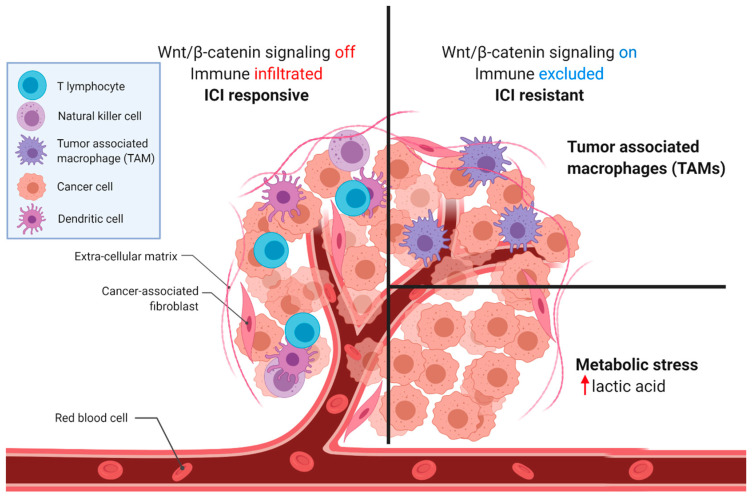
Wnt/β-catenin signaling can alter tumor microenvironment. (Adapted from “Tumor Microenvironment”, by BioRender.com (2020). Retrieved from https://app.biorender.com/biorender-templates).

**Figure 4 cancers-13-00889-f004:**
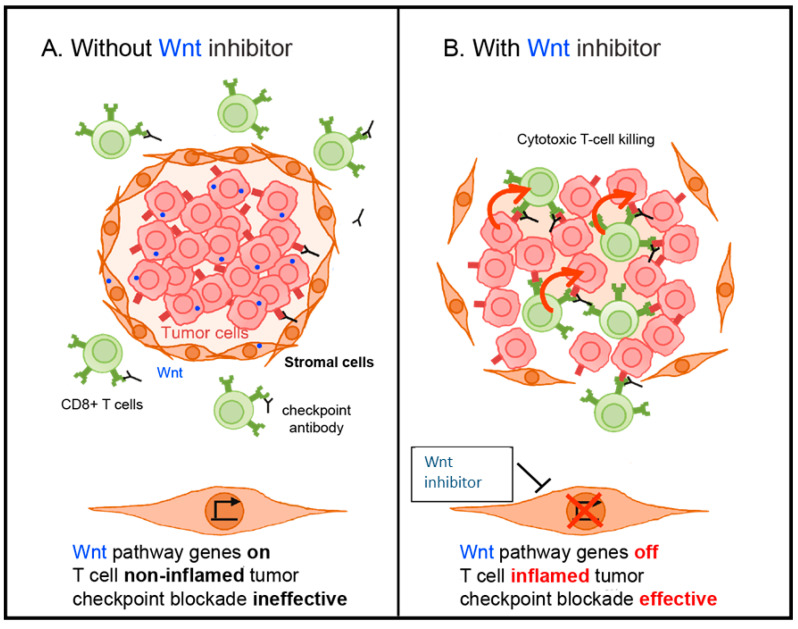
Proposed model for overcoming Wnt signaling driven immune checkpoint inhibitor resistance. (**A**) Immune cell exclusion driven by Wnt signaling. (**B**) Combination of Wnt signaling inhibitor and immune checkpoint inhibitor can overcome resistance.

**Figure 5 cancers-13-00889-f005:**
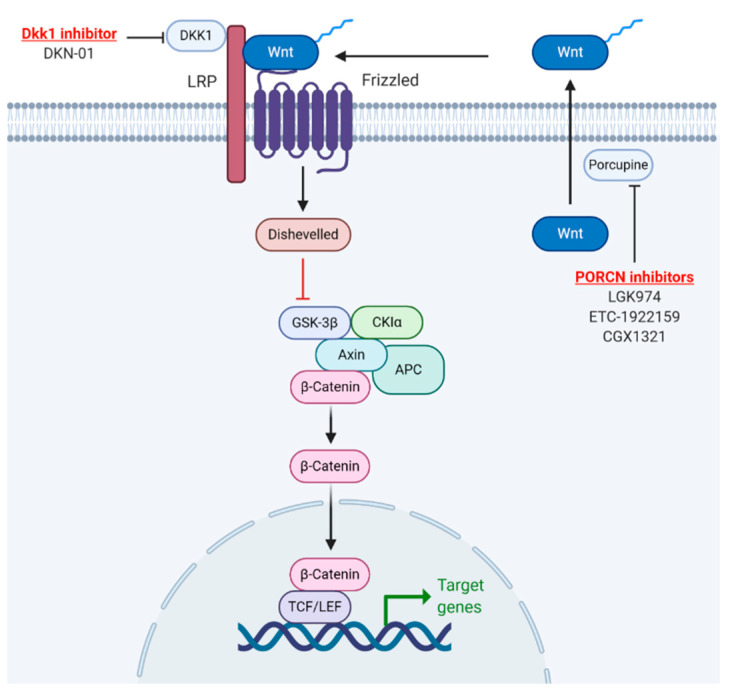
Current WNT/β-catenin inhibitors being used in combination with ICI for human clinical trial. (Adapted from “Wnt//β-catenin signaling”, by BioRender.com (2020). Retrieved from https://app.biorender.com/biorender-templates).

**Table 1 cancers-13-00889-t001:** Current clinical trials combining Wnt inhibitor and immune checkpoint inhibitor.

Drug	ICI Agent	Mechanism of Action of Wnt Inhibitor	Disease	Clinical Trial	Trial Phase
LGK974	PDR001 (anti-PD-1)	PORCN inhibitor	Solid tumors	NCT01351103	Phase I
ETC-1922159	pembrolizumab (anti-PD-1)	PORCN inhibitor	Solid tumors	NCT02521844	Phase IA/B
CGX1321	pembrolizumab (anti-PD-1)	PORCN inhibitor	Advanced GI Tumors	NCT02675946	Phase I/Ib
DKN-01	nivolumab (anti-PD-1)	DKK1 inhibitor	Advanced Biliary Tract Cancer	NCT04057365	Phase II
DKN-01 ± chemotherapy	tislelizumab (anti-PD-1)	DKK1 inhibitor	Advanced Esophagogastric Cancer	NCT04363801	Phase IIa
DKN-01	pembrolizumab (anti-PD-1)	DKK1 inhibitor	Advanced Esophagogastric Cancer	NCT02013154	Phase I

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
