# Peer review of "Wnt/β-Catenin Signaling and Immunotherapy Resistance: Lessons for the Treatment of Urothelial Carcinoma"

_cancers, 2021, doi:10.3390/cancers13040889_

Round 1

Reviewer 1 Report

The present article addresses the highly relevant issue of combining inhibitors of immune checkpoints and of the Wnt/beta-catenin signaling pathway in order to overcome resistance to immunotherapy in urothelial carcinoma. The article is well written and clearly outlines how the data in the literature support this therapeutic strategy. I have no major comments on the text but I strongly recommend that the sections dealing with "attenuation of ICI by CCL4" and "Wnt/beta-catenin induced immune cell exclusion via ETM" be illustrated with schematic figures to improve readability and clarity.

Author Response

We thank the reviewer for their feedback and comments. 

Reviewer comments:

"I have no major comments on the text but I strongly recommend that the sections dealing with "attenuation of ICI by CCL4" and "Wnt/beta-catenin induced immune cell exclusion via ETM" be illustrated with schematic figures to improve readability and clarity."

Response:

Figure 1 - Has been kept in place. 

We have revised the manuscript to include two additional figures to address reviewer #1's comments. 

Figure 2- Provides a schematic for how Wnt signaling attenuates CCL4 and the ICI response.

Figure 3 - This figure is a schematic for how Wnt signaling can drive the inclusion of tumor associate macrophages and metabolic stress (lactic acid production)

Figure 4 - This is the proposed treatment rationale/model for combining Wnt inhibitors and ICI. 

Figure 5 - Current active clinical trials of Wnt inhibitors being combined with ICI in combination for cancer treatment. 

Please see the attached revised version of the manuscript for reference. 

Please see the attached final document for these changes.   

Reviewer 2 Report

In this review the authors cover the WNT/beta-catenin pathway as a mechanism of immunotherapy resistance and therapeutic progress against this pathway. The manuscript is clearly written and no major stylistic or language changes are recommended.  The mechanistic causality is now established between WNT/beta-catenin signaling and immune exclusion in preclinical models, and correlations in clinical tumor specimens has been observed. These data have led to a resurgence in therapeutic development targeting the WNT/beta-catenin pathway with many new agents now in clinical trials. Thus, this manuscript is timely and important. Comments that should be addressed in revision are below.

  1. The review is focused on urothelial carcinoma, though the majority of the evidence and therapeutic trials presented are from other cancers. The authors should elaborate (perhaps in the conclusion) more on why they believe this pathway is particularly important in urothelial carcinoma. Is there other emerging evidence for the role of this pathway specifically in urothelial cancer? How does this pathway fit in with other urothelial carcinoma-specific molecular features (i.e. TCGA luminal/basal/etc subtypes, FGFR3 alterations, etc). Any relevant bladder cancers trials?
  2. The WNT/beta-catenin pathway is complex. It would be helpful to include a figure outlining the various upstream and downstream regulators, especially including the ones targeted therapeutically in Table 1.
  3. Please ensure Table 1 is comprehensive. For instance tehre are a few other relevant drugs in development with trials open (i.e. XNW7201, SM08502, etc)

Minor:

  1. Line 166 "neoantigens" should be singular "neoantigen", as SIY is a single peptide
  2. Line 316: "achieved" should be "who had tumors with" or something similar

Author Response

We thank the reviewer for their feedback and revised the manuscript. Please see our responses below. 

Reviewer comments:

Comments that should be addressed in revision are below.

1. a. The review is focused on urothelial carcinoma, though the majority of the evidence and therapeutic trials presented are from other cancers. The authors should elaborate (perhaps in the conclusion) more on why they believe this pathway is particularly important in urothelial carcinoma. Is there other emerging evidence for the role of this pathway specifically in urothelial cancer? 

-> The reviewer is correct to indicate that most of the studies on Wnt signaling driving ICI resistance has been done on other tumor types besides bladder cancer. As a result we have changed to title to the following:

Wnt/β-catenin signaling and immunotherapy resistance: Lessons for the treatment of urothelial carcinoma.

-> As for the evidence that the Wnt signaling pathway may play a role in urothelial carcinoma carcinogenesis and immune cell exclusion this is discussed on pages 3- 4 (lines 95-153) in the uploaded revised version of the manuscript. We acknowledge in the introduction (lines 57-64) that the majority of the presented work has been done in other tumor types. We also discuss that this is knowledge gap in the field that will need to be addressed directly in the future through the utilization of translational bladder cancer models.  

1.b. How does this pathway fit in with other urothelial carcinoma-specific molecular features (i.e. TCGA luminal/basal/etc subtypes, FGFR3 alterations, etc). Any relevant bladder cancers trials?

-> This is a great question and as far as we know the role of Wnt signaling in relationship to these other urothelial carcinoma molecular features is poorly understood and has not been investigated. This is an area that will need to be investigated in the future and is beyond the scope of this manuscript. 

2.  The WNT/beta-catenin pathway is complex. It would be helpful to include a figure outlining the various upstream and downstream regulators, especially including the ones targeted therapeutically in Table 1.

-> We appreciate the feedback and have added additional figures to the manuscript. In particular this comment is addressed by adding Figure 5. 

3. Please ensure Table 1 is comprehensive. For instance tehre are a few other relevant drugs in development with trials open (i.e. XNW7201, SM08502, etc)

-> The reviewer is correct that there are many clinical trials currently investigating Wnt inhibitors. However, Table 1 is a limited listing of clinical trials investigating the combination of Wnt inhibitor AND immune checkpoint inhibitor. As a result that is why some other inhibitors discussed by the reviewer have not been added to this table. We have changed the Table legend to state "current clinical trials combining Wnt inhibitor and immune checkpoint inhibitor" to make things more clear. 

Minor:

1. Line 166 "neoantigens" should be singular "neoantigen", as SIY is a single peptide

-> We have made this change as recommended. 

2. Line 316: "achieved" should be "who had tumors with" or something similar

-> We have made this change as recommended. 

Please see the uploaded revised manuscript incorporating the reviewers comments. 
